# Cyclicities in the Regime of Groundwater and of Meteorological Factors in the Basin of the Southern Bug River

Oleksii Shevchenko [1,*], Anatolii Skorbun [2,*], Volodymyr Osadchyi [1], Natalia Osadcha [1], Vasyl Grebin [3] and Valeriy Osypov [1]

1 Ukrainian Hydrometeorological Institute NAS of Ukraine, 03028 Kyiv, Ukraine; osad@uhmi.org.ua (V.O.); nosad@uhmi.org.ua (N.O.); valery_osipov@ukr.net (V.O.)

2 Institute for Safety Problems of Nuclear Power Plants of the NAS of Ukraine, 07270 Chornobyl, Ukraine

3 Geografic Faculty, Taras Shevchenko National University of Kyiv, 03022 Kyiv, Ukraine; grebin1964@gmail.com

* Correspondence: shevch62@gmail.com (O.S.); anskorbun@gmail.com (A.S.)

**Abstract:** The data of observations since 1951 in the upper part of the Southern Bug River basin in the west of Ukraine are analyzed. The results indicate that the climate change occurring on Earth disrupts the regular cyclicity of groundwater flow indicators. The identified 7–8-year groundwater level and flow to the river cyclicity correlates well with the air temperature, precipitation and river runoff cyclicity. The noted groundwater cyclicity appears with some delay after the establishment of the 8-year air temperature cyclicity observed since 1969. The manifestation of a 7–8-year cycle depends on the groundwater table (GWT) depth. For shallow groundwater (1.0–2.5 m), such rhythms have been observed since 1975, and for deeper levels, since 1989, which is recognized as the year of the beginning of the climate changes. Moreover, 7–8-year rhythms in the fluctuation of groundwater parameters are characteristic of mainly high-water periods of their multiyear regime, and during the low-flow phase is significantly weakened. During 2011–2014, the groundwater levels abnormally decreased and the 8-year cycles were replaced with 5-year ones. The influence of air temperature on the groundwater regime exceeds the role of other factors. Wavelet analysis was used as the main method of periodicity observation. Gaussian and Morlet wavelets provide the visualization of pronounced periodicities of data. Using multiple correlation analysis, it was confirmed that temperature has become the dominant impact factor on the groundwater (GWT 1.5–4.0 m) regime in recent decades.

**Keywords:** cyclicity; groundwater levels; groundwater regime; temperature; groundwater flow to the river; wavelet analysis





## 1. Introduction

In nature, there are many cyclical processes, ranging from the activity of the Sun and the day–night changes on Earth and the tides in the oceans. Since the terrestrial hydrosphere is an open dynamic system, i.e., it interacts with other systems, its mode is influenced by a large number of external factors. The influence of certain factors is not constant, but subject to the regular course of cosmic and geophysical processes, such as the time of the Earth's rotation around its own axis and around the Sun, the periodicity of maximum approximations and distances of the Earth from the Sun and the Moon. Therefore, in the hydrodynamic regime of hydrosphere objects, many cycles of different durations are observed. Thus, Zaltsberg [1], according to observations in the Leningrad region of the USSR, identified cycles of 3–4, 5–6, 8–14 (average 12) and 26–31 years, emphasizing that the fluctuations of the groundwater table (GWT) are multicyclic. The duration of these rhythms, identified by Williams [2] for precipitation, temperature and runoff variability for much of the world, coincided in many cases with Hale's 10.5-year (21-year) cycles and Glissberg's 88-year solar activity cycles. However, the duration of the cycles of these

indicators in different areas has a large variance in relation to the Hale and Glissberg cycles, so they can only be considered semi-periodic. According to modern views [3,4], the rhythms of climatic indicators, including surface air temperature (SAT) in the northern hemisphere, are controlled by the North Atlantic Oscillations (NAO), variations in the spatial structure of which are determined by Atlantic Multidecadal Oscillation (AMO) in sea surface temperature (SST) [5].

Different factors may have priority in different climatic zones and landscapes [6]. The lower the rhythm frequency, the fewer factors affect the cyclicity and the more clearly it is derived from a particular factor. Conversely, the longer the cycle, the more factors affect it; in addition, at different times, different factors come to be the most dominant, which makes the cycle untenable.

Since global warming over the last three decades has been the most powerful factor influencing the natural environment, it is logical to assume that its impact extends to shallow groundwater aquifers [7–9], and thereby to the regular cyclicality of groundwater.

Universal criteria to prove the impact of global warming on the groundwater regime are still insufficiently developed in Ukraine. It is important to identify clear indicators of this impact, and also to determine the time of its influence at different depths. This is necessary to take into account risks in water use, to assess groundwater recharges and for forecasting.

In our opinion, the cyclicality and patterns in the groundwater regime discovered in the 1950s and 1990s and the forecast models based on them, in the current conditions of climate change and GWT decline, are no longer justified. Therefore, it is expedient, using modern means of information analysis, to study the nature of cyclicality in the groundwater regime and its subordination to regime-forming factors. Given the available literature [10–15] related to the region of Ukraine and, in particular, the Southern Bug River basin, we concluded that the study of cyclicity in the groundwater regime using wavelet analysis has not been conducted to date. Our chosen method (wavelet analysis), in contrast to others, allows us to establish the time of onset of significant changes in the groundwater regime, which are reflected in changes in the cyclicity of GWT and groundwater discharge. The identified patterns in changes in the cyclicity and regime of groundwater runoff can be used to augment the forecast models and flow rates of water intakes. The study of cyclicality in the groundwater regime for other watersheds of Ukraine will allow us to establish patterns related to climatic zonation and the landscape timing of GWT, which will allow us to make more accurate forecasts and improve the rational management of groundwater resources.

The aim of the work is to test the assumption that global warming is becoming the most influential factor on the shallow groundwater regime and the cyclicity of its parameters. To do this, it is necessary to study the nature of changes in cyclical fluctuations in groundwater levels and their runoff into the river in the periods before and after the onset of abrupt climate change, and to investigate the relationship of these changes with the change of temperature for the possibility of groundwater regime forecasting.

## 2. Research Methods

To test the working hypothesis and to determine the degree of connection between the GWT regime and climate change, we compared the cycles in the regime of groundwater levels and flows with the cycles of temperature and precipitation. Wavelet analysis was chosen as a modern statistical method for allocating cyclicity in the series of data. Wavelet analysis is very convenient for searching for periodic components in long, regular series of measurements. Moreover, its ability to identify such components in individual parts of the series (individual time intervals of measurements) gives it an advantage over the Fourier transform. It is this feature that makes it possible to conduct an analysis, the results of which are given in the text of the article, and to determine the time intervals when there is a change in the periodicity of processes. The method is widely described—a simplified statement can be found in [16,17].

Different mother wavelets can be used for finding different peculiarities in signals. Figure 1 demonstrates the possibilities of wavelet analysis, based on Gaussian wavelets, in

terms of finding periodicities in the signal. The picture shows a signal in the form of a sine wave of a certain frequency and a two-dimensional matrix of coefficients of the wavelet decomposition in the form of an image, where the color or shade of gray shows the value of the coefficients. The time in the form of a number of set elements is shown horizontally (measurements are carried out regularly in a certain interval of time). The half-period of a sinusoid in units of a horizontal axis (scale) is shown vertically. In a simplified explanation, the value of the wavelet decomposition coefficient is proportional to the correlation between the time-limited function called a wavelet, and the corresponding part of a series of data.

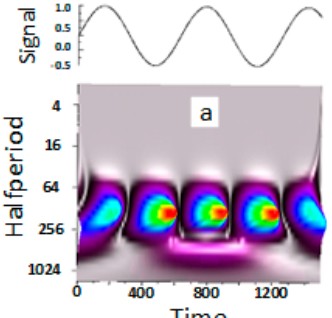 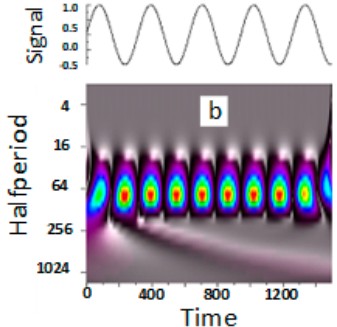 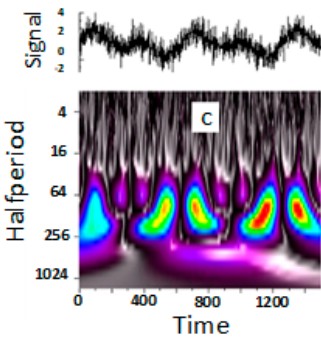

**Figure 1.** Examples of wavelet-periodograms with obvious periodicity in data distribution: (**a**)—the signal in the form of a sinusoid and the result of its wavelet transformation using a Gaussian wavelet; (**b**)—result when the frequency of the sinusoid is twice as high; (**c**)—the signal is the sum of two previous sinusoids with the adding of Gaussian noise.

Figure 1a shows a signal in the form of a sinusoid of a certain frequency and the result of its wavelet transformation using a Gaussian wavelet. It is clearly seen that the wavelet transform looks like a series of horizontally arranged spots. The distance between the spots is equal to the half-cycle of the sinusoid (each spot corresponds to the maximum or minimum of the sinusoid). Figure 1b shows the result when the frequency of the sinusoid is twice as high: there are also a number of spots that are located slightly higher. This corresponds to the fact that the period of the sinusoid has become shorter. Finally, in Figure 1c, the decomposition result is shown for a situation where the signal is the sum of two previous sinusoids with the adding of Gaussian noise of appropriate intensity. The picture of its wavelet transformation below shows rows of spots from these two sinusoids, although distorted by noise. This picture also gives an idea about the sensitivity of the method relative to the signal-to-noise value.

If a long dataset includes a pronounced periodic component, then it will appear as a series of regularly located horizontal spots of higher intensity in the interval of this periodicity.

Therefore, when searching for periodic changes (possibly hidden in the noise) in the analyzed signal using this approach, in the picture of wavelet coefficients, you need to look for horizontally arranged rows of spots.

The method is, to some extent, qualitative in the sense that it works reliably in situations where the presence of a set of spots is determined unmistakably, albeit "by eye".

A good example of this method's sensitivity is regular measurements of a temperature (see below): a sinusoid, produced by seasonal temperature changing. This sinusoid is slightly distorted due to the influence of multiyear meteorological changes. The wavelet transformation of this function is shown in the squared image as a system of spots. The most prominent feature in this image is a light horizontal band (marked by an arrow) on the half-period—about 6 months—which, in reality, consists of the series of up-extended spots. Each of these spots corresponds to the maximum or minimum of an upper sinusoid, that is, to winter or summer temperatures. Less prominent, but still visible, is the series of rounded spots on a half-period of about 48 months, which corresponds to about an 8-year periodicity.

Such peculiarities, i.e., series of regularly horizontally positioned "spots" on different (half) periods is the subject of our analyses. Due to the fact that the existence of such regular

patterns is a qualitative manifestation of the existence of periodical components in the signal, we name our wavelet images "wavelet-periodograms" for short. With sufficiently long measurements, it may be that the periodic process began at some point in time and ended after some time (not lasting for the full duration of the measurements).

The main feature of wavelet analysis for us is that, in the picture of the wavelet coefficients, the corresponding spots will only be present in this time interval (unlike the Fourier transform, which only gives information about the presence of the periodic component without linking it to the time of appearance). As for the uncertainty of the estimate of the time of the spot's appearance, this is determined by the accuracy with which one can determine the center of the spot. Accordingly, it can be assumed that the uncertainty is approximately half the width of the spot in units of the horizontal axis. Thus, the accuracy of the estimates is determined by the accuracy of finding the center of the spot and is mostly $\pm (2 \div 3)$ units of the horizontal axis.

## 3. Characteristics of the Object and Factors Influencing the Groundwater Regime

To identify cyclical fluctuations in groundwater levels and flow into the rivers of Ukraine, we selected a series of one of the longest (since 1951) observations of GWT in groundwater wells in the Southern Bug River basin (Vinnytsia region), the runoff of which originates in the zone of sufficient moisture, and is provided in a zone of insufficient humidification. This is the only river basin that is completely located within the territory of Ukraine (Figure 2).

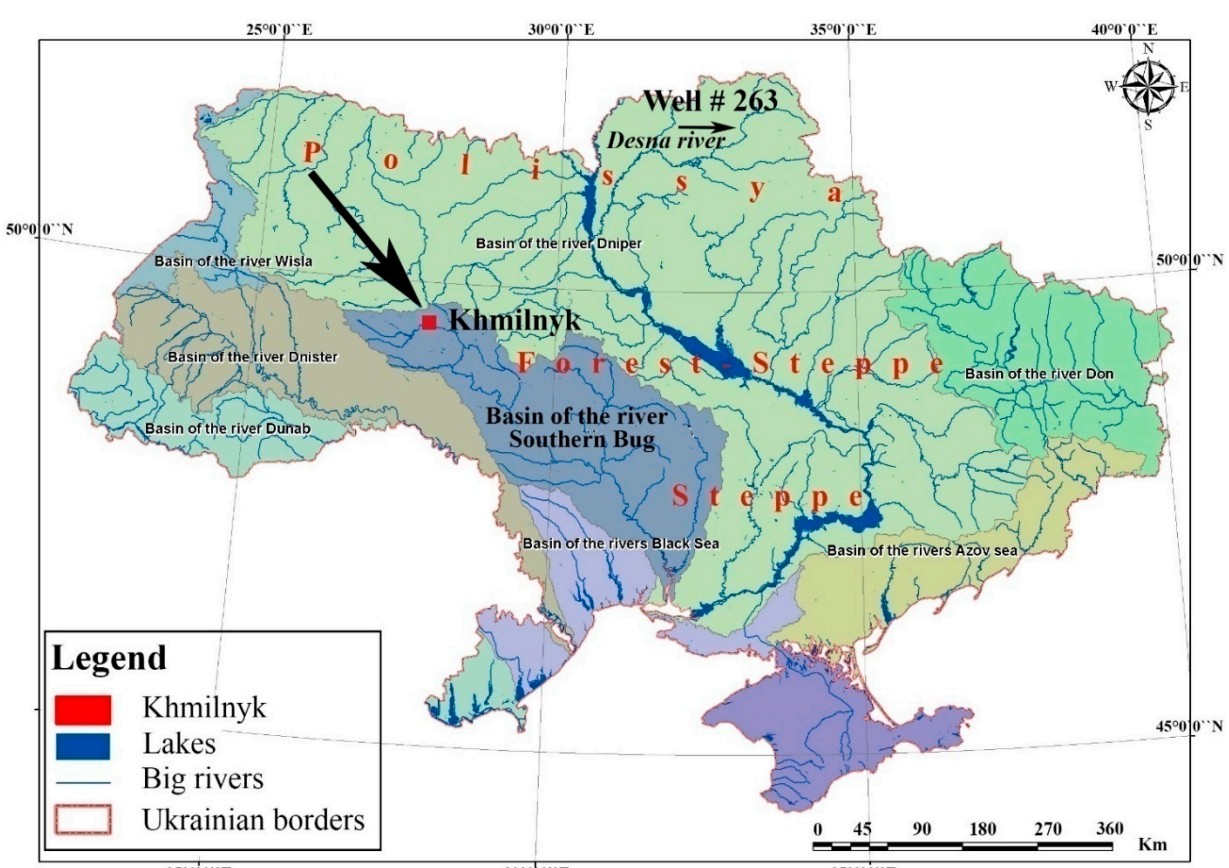

**Figure 2.** Map of the main river basins within Ukraine. Red square denotes location of experimental hydrogeological control section (wells #5-2, #5-3, #5-5, the river gauge station), a small black arrow indicates the observation well №263 on the Desna River.

Until recently, we could observe an inverse dependence of GWT on solar activity. The Sun was most active in April 2014: the peak corresponded to, on average, 82 sunspots. Due

to abnormally low annual precipitation (40% below normal) in the territory of Ukraine in 2015, the minimum GWT coincided with the maxima of solar activity, which was recorded during the 24th solar cycle. This year was not marked by an increase in the intake of fresh surface and groundwater (Figure 3).

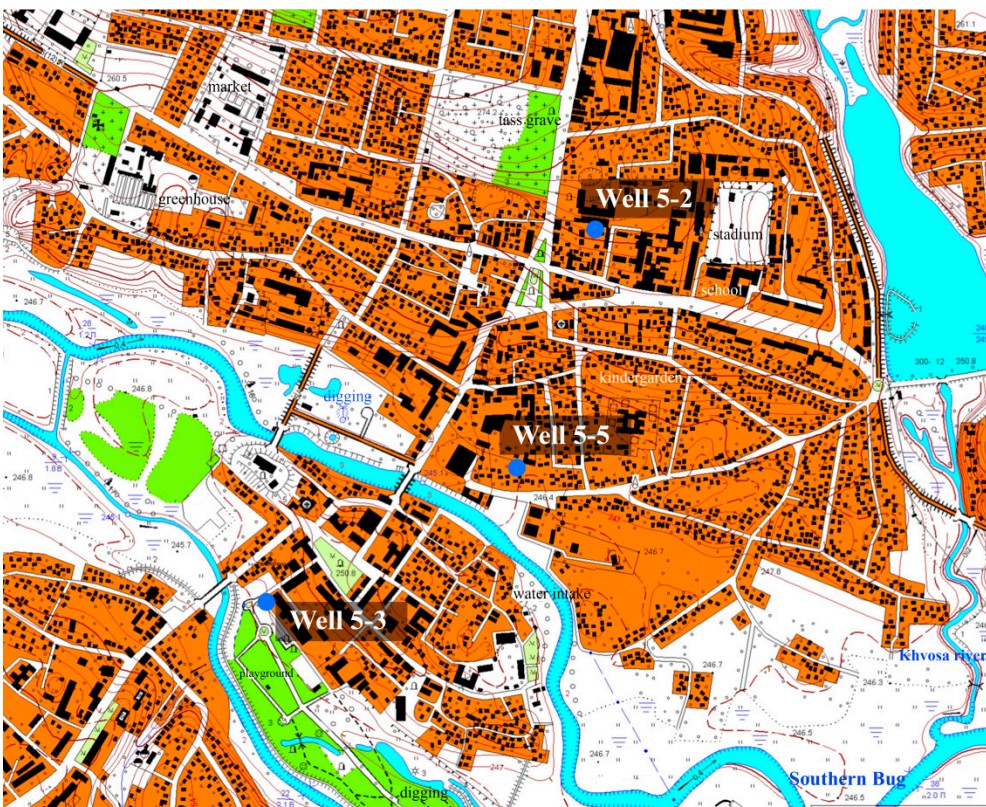

**Figure 3.** Location of wells #5-2, #5-3 and #5-5 within the city of Khmilnyk.

The period of the lowest activity of the Sun during the 24th solar cycle, which recently ended in the recession phase, lasted from August 2019 to September 2020 [18]. Contrary to this pattern, 2019 and winter 2019–2020 were marked by a decrease of GWT on 0.5–0.8 m below the long-term average, and the precipitation in the spring–summer of 2020 did not contribute to the full recovery of levels. This indicates a violation of the natural subordination of hydrogeological events and processes, which may be due to radical changes in the nature of recharge (recovery of groundwater) under the influence of modern climate change.

The current air temperature increase in Ukraine is still growing and, obviously, the associated climate change must affect both the surface and underground parts of the hydrosphere.

Weather conditions. Observations include contrasting periods of thermal regime and surface atmospheric pressure. The period 1951–1980 was characterized by reduced pressure in most of Ukraine, which was accompanied by climate change. In summer, there were long rains and relatively low temperatures. After 1980 and until about 1997, the average field of the surface atmospheric pressure in winter was determined by the impact on Ukraine's subtropical maximum with warmer winters and precipitation, mainly in the form of wet snow and rain. Between 1989 and 1998, there was a significant positive anomaly of average monthly winter temperatures. At the same time, short-term penetrations of Arctic cold air were observed, which led to sharp changes in the average daily air temperature and were accompanied by a significant increase in wind.

Near Khmilnyk city, there are, on average, 40–45 winter days with thaw, of which 40% fell in December. In January, the number of thaws are the lowest. In the river valleys, the number of days with thaws exceeded their number in watersheds (plains). Since 1987, the number of days with thaws has exceeded the long-term norm. There is no clear periodicity

in the appearance of the warmest winters. Such winters were in 1960–1961, 1965–1966, 1974–1975, 1982–1983, 1988–1989, 1989–1990, 1993–1994, and others. Low-snow and cold winters with liquid thaws also do not have a clear periodicity.

The weather conditions of the warm season have also become more volatile.

Water balance. The main balance district is confined to the upper part of the Southern Bug River and is located within the city of Khmilnyk, Vinnytsia region (Figure 2). The catchment area of the Southern Bug River to the city of Khmilnyk is about 4.250 km$^2$.

The historical average annual precipitation near Khmilnyk is 635 mm, 40.6% of which falls in summer. The total evaporation averages 555–561 mm year$^{-1}$, and by seasons it is distributed as follows: 15–40 mm in winter; 158 mm in spring, 305–310 in summer and 77 mm in autumn [19]. The annual recharge of the groundwater was 238 mm until 1988; later, it fell to 124 mm year$^{-1}$. For GWT fluctuations within the range from 2.5 to 4.5 m depth, infiltration recharge on the hillslope of the river valley averaged 18.5 mm in the period 1980–1988 and 14.8 mm in the period 1989–2017.

Near the village of Lelitka, 1 km upstream of Khmilnyk city, the multiyear annual runoff averaged 470 million m$^3$ year$^{-1}$ during 1980–1988, and later (2011–2020) decreased to 258.6 million m$^3$ year$^{-1}$. The water abstraction for the drinking supply of the city of Khmilnyk varied from 0.55% to 0.59% of the annual runoff of the Southern Bug River. Within the Vinnytsia region, the rivers are fed by 48% rainwater, 25% snow and 27% groundwater [20].

The mean surface runoff of the Southern Bug River basin near Khmilnyk city was 13.3 m$^3$ s$^{-1}$ (419.43 million m$^3$ year$^{-1}$) or 98.7 mm for 1980–2019. The specific groundwater flow of 95% probability is 0.24–0.3l s$^{-1}$ km$^{-2}$ in the small basins of various tributaries. According to our estimations, the long-term average underground flow of 50% probability was 146.9 million m$^3$ year$^{-1}$, which corresponds to 34.56 mm and 35% of the annual surface runoff. To obtain these values, the groundwater discharge was calculated for the minimum surface runoff (summer or winter) of rivers in the years of 95%, 85% and 75% probability, and then by linear extrapolation—groundwater flow of 50% probability. The share of underground flow increased by 6.4% compared to the period 1964–1988.

The highest groundwater discharge into rivers is observed in summer. This season exhibited the long-term maximums of daily and monthly groundwater flow.

Groundwater. The study area includes three observation wells, two of which (#5-3 and #5-5) open the first free-horizon groundwater level in alluvial Upper Neo-Pleistocene and Holocene deposits, and the third (#5-2) is the second from the surface, low-pressure horizon (pressure 2.8 m above the confining bed) (Figure 3).

Well #5-3 is located on an island, in a floodplain, at a distance of 84 m from the river, with well #5-5—on the first left-bank terrace—136 m from the river. The groundwater regime in the area of the first well should be recognized as riverside, and in the second well as the valley side (transit). This is confirmed by the degree of correlation between the groundwater and surface water levels. Thus, for well #5-3, the correlation coefficient was 0.56 by 1998, and later 0.4, and for well #5-5, the correlation coefficients were, respectively, 0.4 and 0.26. The correlation between groundwater flow and surface runoff was mostly insignificant. At the first site, the correlation coefficients changed from 0.44 in the period before 1988 to negative values at a later time. At the second site, the highest correlation coefficients reached the value of 0.60–0.65 and were observed in the summer season from 1980 to 1998. More than 10 km downstream is a small reservoir 2.4 km long. The backwater zone of this reservoir does not reach the well even at maximum levels. Irrigation of the nearby agricultural land is not carried out. On the island, along the shoreline where well #5-3 is located, there is a road with a low embankment. It serves as a shore protection dam. This sometimes creates an obstacle to surface runoff during snowmelt and keeps the higher levels longer at the well.

Five radon mineral water wells with depths from 83 to 100 m are operated in the area of well #5-3. The aquifer of mineral waters is protected by a confining layer up to 11 m. During 2010–2016, mineral water uptake ranged from 145 to 215 m$^3$ day$^{-1}$ and

caused the fall in the water level of 0.6–24.9 m in the wells. Steady piezometric levels in groundwater wells are set at a depth of 1.8–5.1 m. The mineral water withdrawal does not have a significant impact on the regime of groundwater in well #5-3. Groundwater recharge is possible in the area of well #5-3 from a confined aquifer in some seasons of the year during the decrease of GWT.

The water supply to Khmilnyk is provided by both the surface water of the Southern Bug River and ground water. Among them, surface water dominates. Surface water abstraction equals 3.5 thousand m³ day⁻¹ and is located 3.0 km upstream from the center of the island. Groundwater intake is located 3.7 km downstream of the river, which eliminates its impact on the groundwater regime in well #5-5. Figure 4 shows the annual dynamics of the water supply of Khmilnyk from different sources. The dynamics of water abstraction does not have any predictable regularity or cyclicity.

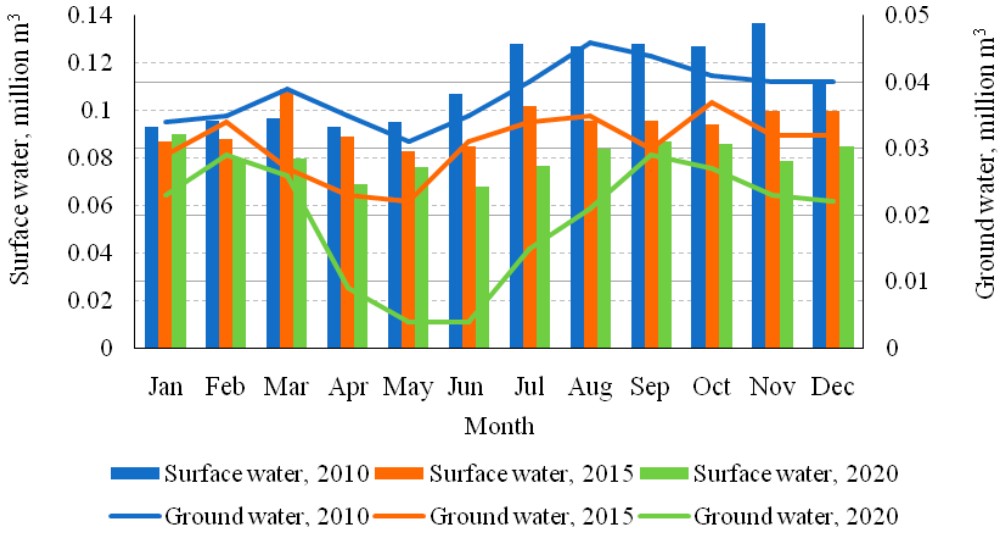

**Figure 4.** Dynamics of monthly water abstraction from surface water (the Southern Bug River) and deeper groundwater (wells) for Khmilnyk's drinking water supply.

Based on the factors mentioned above, the groundwater regime in well #5-5 can be recognized as undisturbed.

Water-bearing rocks in the geological section of well #5-3 are represented by yellow inequigranular sands with layers sandy loams, while in well #5-5, coarse-grained sands predominate. Well #5-2 is located at a distance of 730 m from the river and opens the horizon in the Paleozoic–Mesozoic crust of weathering of crystalline rocks (granite gruss, rock debris, fine sand).

## 4. Research Results

Cyclicity of GWT. For the groundwater regime until 1986, the sustained recurrence is observed for low-water years of 82–99% supply with the periodicity of 4–6 and 10–11 years, which are consistent with the cycles of solar activity [21]. However, there is no such recurrence after 1986. The period of high GWT, which began for the aquifer in the Upper Quaternary and modern alluvial deposits (well #5-5) after 1989 (Figure 5), corresponds well with the first transition of average monthly temperatures in February to the positive values and relatively stable maintenance of average temperatures in February above −4.5 °C [13]. In general, 1989 was recognized as the year of "the beginning of significant climate change in Ukraine" [22]. In previous publications [21], we proved that it was the winter thaws, which have sharply increased since 1989, that led to an increase of infiltration supply, a rise in GWT, and large-scale flooding in Ukraine. Thus, certain meteorological "markers" also correspond over time with the changes in the groundwater regime.

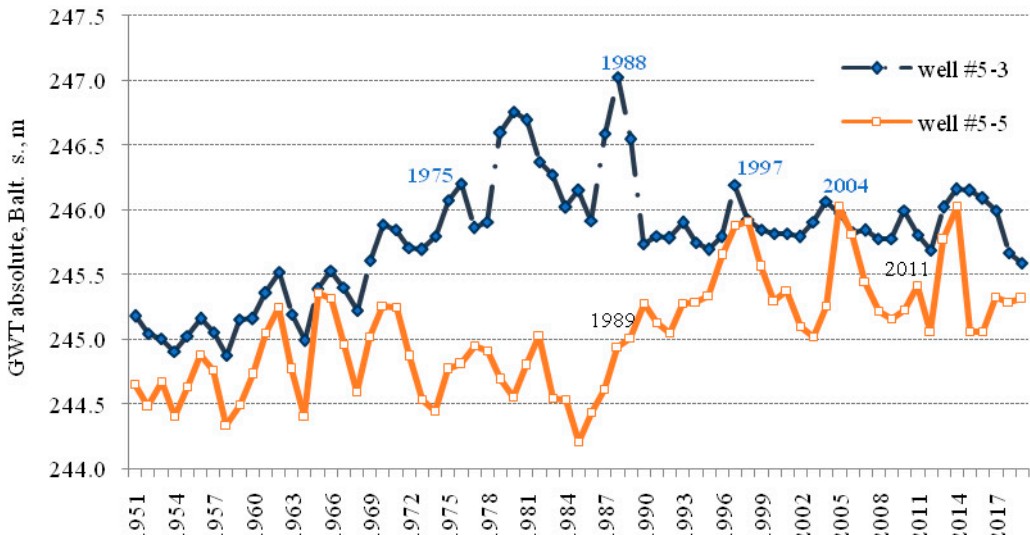

**Figure 5.** Chronological graphs of GWT in the wells: #5-3 (GWT = 1.0–2.5 m) and#5-5 (GWT = 2.5–4.5 m), aquifer in alluvial Upper Neo-Pleistocene and Holocene sediments of the first flood-plain terrace, Khmilnyk, Vinnytsia region, Ukraine.

From the graph, it can be seen that the increasing difference in the depth of GWT disturbed the synchronicity of the fluctuation levels in both wells; with the deepening of the level in well #5-5 after 1971, the delay time of the extremums (precipitation response) increases.

According to the results of the wavelet analysis of the monthly averaged values of GWT, we found a 7–8-year cycle of level fluctuations for all observed wells with different GWT, with an intact or slightly disturbed regime in the upper part of the basin of the Southern Bug River (Vinnytsia region). In particular, for well #5-5 (Khmilnyk), with an average long-term GWT equal to 3.65 m, such cyclicity is most clearly observed from about 1989 (Figure 6). At the same time, the 10–11-year cycle corresponds to a half-period of about 64 months in the monthly wavelet-periodogram, which is very indistinct. It can only be seen in a large scale as 4–5 spots, which correspond to the low-water period of 1972–1986. This short-term manifestation of the 11-year cycle almost fits into the 21st solar cycle of 1976–1986. Pekárová [4] noted that water cycles of 44, 22 and 11 years are usually associated with solar activity, while longer cycles of 28–31 years are most often associated with North Atlantic Oscillation (NAO).

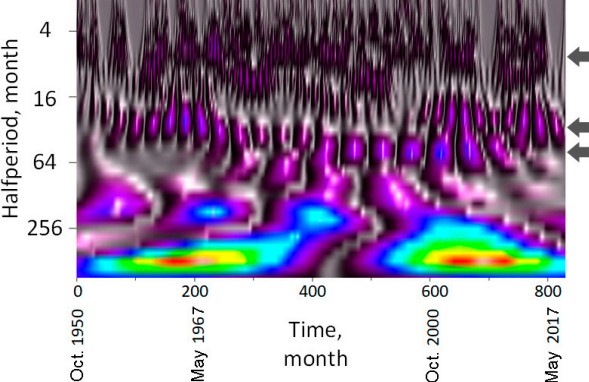

**Figure 6.** Wavelet-periodogram for GWT in the well # 5-5 (Khmilnyk, left bank of the Southern Bug) for 1950–2020. The horizontal axis represents the observation time in months; the vertical axis represents the half-period in months. The arrows show a series of spots, allowing us to reliably determine the cyclicity period: top arrow—1 year, middle arrow—5–6 years and lower arrow—8 years. The colors in the order of increasing intensity reflect the corresponding values of the wavelet transform coefficients.

According to the wavelet-periodogram of a fairly long series of observations of GWT (Figure 6), approaching 70 years (832 months), we can see that at the beginning, there is a 5–6-year cyclicity, but it is not constant. It is seen that spots of its half-cycles first arise, possibly changing the 7–8-year cycle, and then, after about 220 months, gradually move lower, starting a series of 8-year cycles. The latter also lasts for a short time and practically disappears after 2011, which can be explained by a significant increase in summer temperatures and the reaction of the GWT regime to the beginning of the low-water period in 2008–2009. We emphasize that the beginning (1989) and the end of the 8-year cycle do not coincide with the beginning and end of the solar cycles (1986–22nd, 2009–23rd), but its total traced duration, about 22 years, is multiple for the solar cycles' duration. In our case, 5–6-year cyclicity is typical for long stages of low GWT, and 7–8 years for high-water periods of high GWT.

On the wavelet-periodogram of the GWT chronological series (depth 1.0–2.5 m from the surface) in the well #5-3 (Figure 7), located on the island in the middle of the Southern Bug River in Khmilnyk, as in the previous case, from the beginning of observations in 1951, there is a noticeable formation of rhythms averaging 5 ± 0.6 years (creating a horizontal row between marks 16 and 64 on the y-axis), and rhythms at 8.1 ± 1.0 years begin be most clearly traced from about 1974–1975, i.e., somewhat earlier than in the previous case, due to the earlier rise of GWT in this area from 2.23 m on average for 1951–1974, to 1.5 m for the period 1975–2019. With the advent of 8-year cycles, 5–6 year "solar" half-cycles begin to become "lost" and are eventually indistinct. This may mean that the new 7–8-year rhythm is caused by a more influential global factor than solar activity, because the actual 8-year cycles for the latter are not typical. However, after 2004 (503 months on the abscissa axis), these rhythms have almost disappeared, despite the lack of significant changes in the GWT regime in this area.

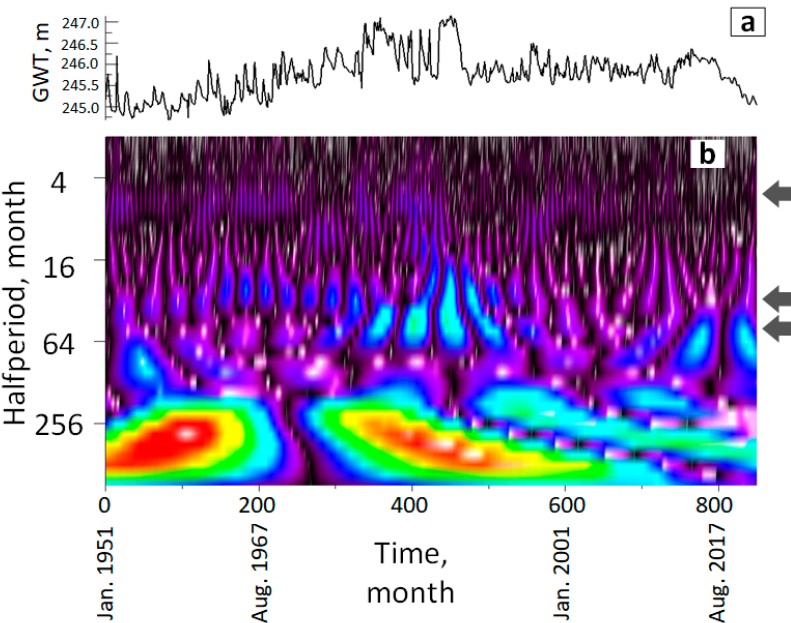

**Figure 7.** Top chart (**a**) is a chronological series of average monthly values of GWT (BSA, m) in the well # 5-3 (Khmilnyk, island part, 1.0–2.5 m) for 1951-2021. Bottom figure (**b**) is a two-dimensional picture of the coefficients of GWT wavelet-periodogram. See details in Figure 6. Cyclicity periods: top arrow—1 year, the middle arrow—5–6 years, lower arrow—8 years.

For the groundwater at the catchments of the tributaries of the Southern Bug River to also have close to 7-year cycles with a slightly shorter duration is typical. Thus, in the basin of the Sob River (city of Lypovets) for GWT (3.0–6.0 m), from the beginning of observations in 1988, cycles of 6.9 ± 0.3 years were established, but in 2013, they disappeared. For GWT

(0.4–1.8 m) in the upper reaches of the regulated river Zgar (village of Horodyshche), the finding more or less clearly shows the cyclicity at 7 years (observation began in 1986).

However, for confined waters, the level of which fluctuates below 13.0 m from the surface (well #5-2, Khmilnyk), the 7–8-year cycles are very weak. The groundwater regime of the second water-bearing horizon from the surface is characterized by cycles of 4.4, 12.5 and 42 years. The cyclicality of 4.4 years is not constant and has a gap from 1998 to 2010—just when the 7.7-year cyclicality arises and intensifies. However, for GWT less than 2.5 m, such cyclicity is weak. It is probable that the cyclicity of 7–8 years has limitations according to the depth of the groundwater, which confirms the external nature of the factor. Additional research is needed to confirm this.

It should be noted that in the history of regime observations of GWT, 6–8-year cycles have already been distinguished, particularly when processing data on temperature fluctuations, precipitation, minimum runoff of the Don River, which reflects the groundwater flow to the river, and GWT on September 1 each year in Kamennaya Steppe (Voronezh region of the Russian Federation, left bank of the Don River) [23]. In addition to the 7-year cycle, other cycles were also manifested, but the clearest were 11 and 7 years. Seven-year stable cycles are also quite clear for the flow of the Dniester and Prut rivers, along with 5- and 29-year cycles, which go beyond random origin [24]. Moreover, 3.2- and 6–8-year cycles in the growing season air temperature were revealed by [25] using tree-rings of European beech.

Cases of the "disappearance" of the 11-year cycle in the dynamics of various hydrological phenomena have also been observed (particularly for the levels of Lake Victoria and the Caspian Sea), that [26] explained by the "hiding" of them by clearer 5–6-year cycles.

It would seem that constant and common to the entire water shell of the Earth, space objects and geophysical processes should form if not synchronous cycles, then at least rhythms with the same repetition. However, it has turned out that there is not enough sustained recurrence (cyclicity) in the changes of the hydrology regime. Not only are rhythms of different periodicity observed, but also the periodic disappearances of certain rhythms, the appearance of other cycles, and different rhythmicity for the groundwater of different catchment basins. The latter can be caused not only by zonal climatic factors [22], but also by landscape differences [6].

As mentioned above, topography, geomorphological features and GWT determine the conditions for groundwater recharge and have a significant impact on their cyclicity. These factors exceed the role of species composition and the nature of vegetation distribution in the research place (evapotranspiration is lower than the average zonal value: see above). Important factors affecting the evaporation and recharge of groundwater are the lack of air humidity as well as the direction and strength of the wind. Atmospheric fronts, along with other factors, generate static electric fields of the surface atmosphere, which have a significant impact on the movement of moisture in the aeration zone and the regime of subsoil water [27,28]. Thus, cold fronts bring predominantly negative charges, which lead to a decrease in GWT, and warm fronts bring positive discharges, which contribute to an increase in GWT.

It is important to note that in the so-called transition period to the 24th solar cycle (2008–2009), violations in the synchronous changes of the 11-year cycle of solar activity and global temperature, which were observed up to 1998, were also detected [29].

There is reason to believe that profound, radical changes in the groundwater regime have taken place in recent decades.

To confirm that the 8-year cycle is a "marker" of changes in the groundwater regime in the first phase of global warming, we investigated GWT rhythms in the Desna River basin (Figure 8). They are clearly visible in areas with GWT from 5 to 11 m for the same period as in the basin of the Southern Bug River, from 1989 to 2011. Such rather clear rhythms show that in the territories located to the north (Polissya or Woodland), the 8-year cycle is manifested better and to greater depths of GWT. The first phase of warming due to rising winter temperatures and increasing numbers of thaws resulted in a significant increase

of infiltration groundwater feed [21]. Polissya has more pronounced climate changes because of higher winter precipitation, lower temperatures and smaller GWT depth, which is likely to contribute more to climate change appearance. Since 2011, the second phase of warming has begun. A hydrogeological drought accompanied by a decrease in the GWT was observed everywhere in Ukraine. In Polissya, water disappeared from rural wells.

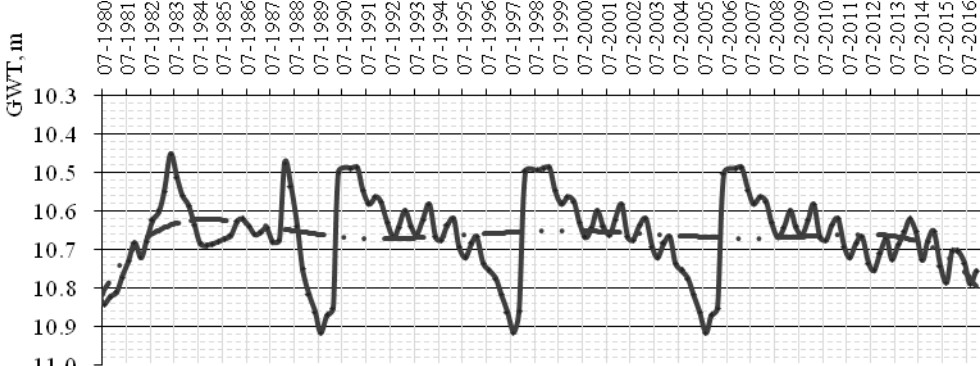

**Figure 8.** Fluctuations of seasonal average GWT (well #263) in the Desna River basin (a tributary of the Dnipro River basin) with a well-defined 8-year cycle.

Cyclicity of groundwater flow. 7–8-year cycles are clearly visible on the wavelet-periodogram of rate of groundwater discharge to the Southern Bug River (Figure 9), which was calculated numerically [30] according to hydrogeological and hydrometric monitoring observations. The chronological graph also shows two periods in the nature of changes in the underground inflow to the river: (1) monotonous and rapid growth (1980–1998); (2) balancing of the general tendency (1999–2019), but with the available sharp extremes—the highest maxima and rather deep minima. Recurrence of maxima—8–9 years.

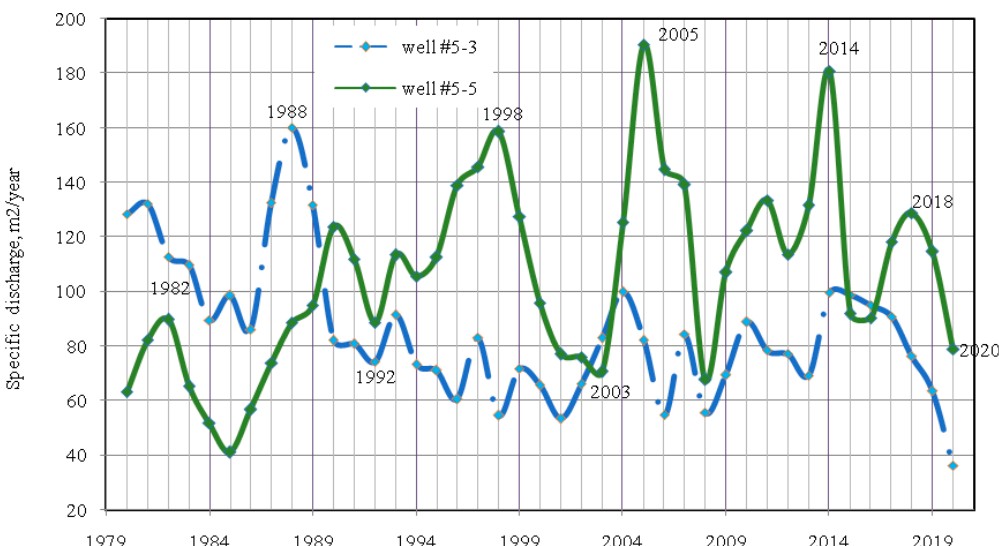

**Figure 9.** Chronological graph of variability of values of specific rate of groundwater discharge to the Southern Bug River, m$^3$ year$^{-1}$ per running meter of the length of the river for the period 1980–2017 in the range of the Khmilnyk city, with the allocation of maxima having a 5–6 and 10–11 periodicity for well #5-3 and 7–8-year frequency for well #5-5.

From the given chronological graphs of groundwater flow, it is clearly visible that various conditions, including GWT and area of catchment, affect fluctuations, amplitude and the formation of runoff extremes. However, a pronounced 8-year cyclicity was found for both cases. The higher the GWT, the earlier this cyclicity begins to manifest.

The boundary between these periods is also substantiated by the time of disturbance of the synchronicity between solar cyclicity and global temperature. It is typical in the first period for a tendency of growth in the numbers of days with intense 24-h precipitation and long-term 1–7-day precipitation, the sum of which exceeds the threshold for the weather station values [31]. Together with the pointed winter thaws, which become more frequent from 1987 to 1989, this contributed to an increase in groundwater resources. On the island site with GWT 1.0–2.0 m (well #5-3), the maximum discharge of groundwater to the river falls between 1987 and 1989 (Figure 9). A yearly sum of precipitation tends to grow insignificantly [21]. The noticeable irregularity of the groundwater flow during the second period is stipulated by the uneven distribution of monthly precipitation, which changed without a clear tendency. The yearly sum of precipitation is thereby decreased, a trend that is noticeable in many countries of the world [32]. Over the area with high GWT (well #5-3), over a minimal period, the entirety of observations of groundwater flow to the river began in 1998.

The largest volumes of groundwater flow occur in the summer months, when the water level of the river is lowest. At around 2013–2014, the long-term trend of balanced supply and unloading of groundwater in the Forest–Steppe zone with a predominant infiltration-washing regime came to an end. Prior to this, the unloading of groundwater into the rivers of the basin of the Southern Bug River was completely compensated by infiltration feeding [21]. Since 2015, groundwater levels throughout the country have entered into a recession. We can expect a gradual reduction in groundwater runoff in catchment areas with GWT > 2.5 m [33].

According to the results of the wavelet analysis of groundwater flow data to the Southern Bug River (Figure 10), a clear 7.5-year cycle is confirmed, but has become more obvious since 1993 (half the rhythm after 1989).

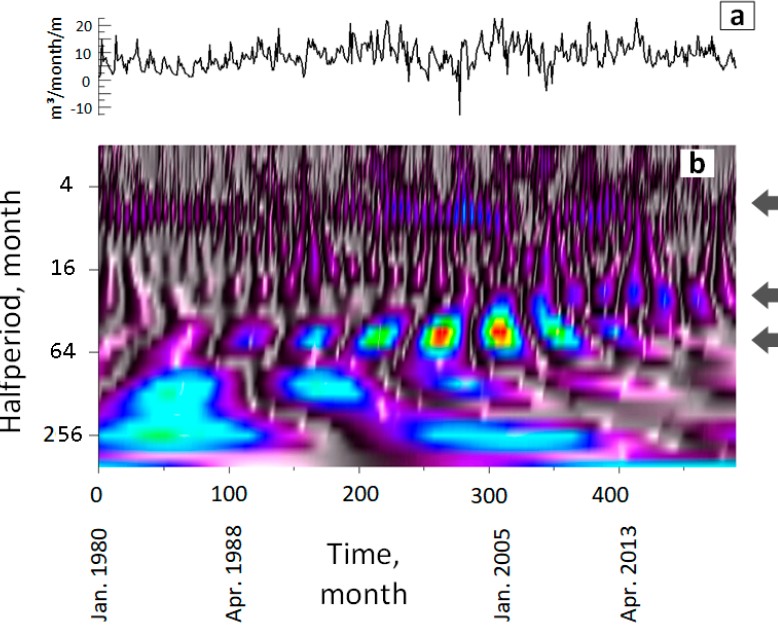

**Figure 10.** Wavelet-periodogram (**b**), based on the results of the calculated data (1980-2020) of the underground discharge (m$^3$ month$^{-1}$ m$^{-1}$) to the Southern Bug River (**a**). Distance between spots corresponds to half of the cycle. Cyclicity periods: top arrow—1 year, middle arrow—5–6 years, lower arrow—8 years.

In the wavelet-periodogram of groundwater flow, in contrast to GWT (see Figures 5–7), 5–6-year rhythms are almost never observed. However, according to the behavior of the appearance of spots of 7–8-year cyclicity, and according to Table 1, it is clearly seen that the

duration of the cycle increases from 5.8 to 8.0–8.5 years over me. It is logical to assume that the 7–8-year cycle of groundwater flow, as well as of GWT, was preceded by a 5–6-year cycle.

**Table 1.** Description of rhythms, which are exhibited in the wavelet-periodogram of the underground flow to the Southern Bug River.

| Spot Position on a Horizontal Axis, Month | Interval, Months | Period, Months | Period, Years | Calendar Year |
|---|---|---|---|---|
| 1 | 2 | 3 | 4 | 5 |
| 6 | | | | |
| 39 | 33 | | | |
| 76 | 37 | 70 | 5.8 | 1985 |
| 117 | 41 | 78 | 6.5 | 1989 |
| 165 | 48 | 89 | 7.4 | 1993 |
| 214 | 49 | 97 | 8.1 | 1997 |
| 259 | 45 | 94 | 7.8 | 2002 |
| 311 | 52 | 97 | 8.1 | 2005 |
| 361 | 50 | 102 | 8.5 | 2009 |
| 407 | 46 | 96 | 8.0 | 2013 |

Spectral analysis of the data by the method of fast Fourier transform confirmed the cyclicity of 7.4 years (88.8 months) for the monthly specific (m$^3$/running m/month) of groundwater flow, as well as cycles of 1 and 4 years [21].

Cyclical surface (river) runoff. If 7–8-year cycles in the GWT regime are determined by general climatic factors, and not by hydrogeological conditions or local drainage conditions, then a priori they should be manifested not only in the groundwater regime, but also in the surface waters. Indeed, in fluctuations of the Southern Bug River discharges there are 7–8-year cycles, but only up to and including 2003 (Figure 11). Later, they almost disappear, but instead, from 2004, there are cycles lasting 11–12 years and from 2008 (the beginning of the low-water cycle in the surface water regime) there are 5–6-year cycles.

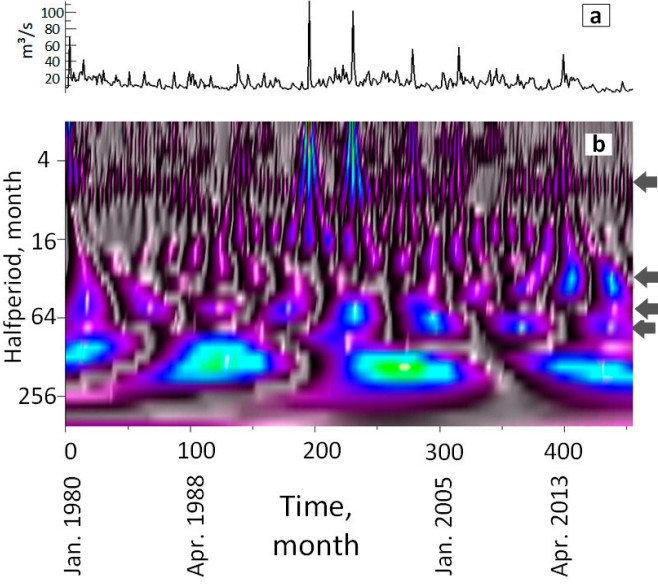

**Figure 11.** Wavelet-periodogram (**b**) based on the average monthly values of the Southern Bug River discharges (m$^3$ s$^{-1}$) for the period 1980–2017 at the post in the village of Lelitka (**a**). The arrows show a series of spots, allowing us to reliably determine the cyclicity period: top arrow—1 year, middle arrows—6 and 8 years, lower arrow—11 years.

Manifestations of 5–6 annual cycles provide a basis for a meaningful analysis of the impact of the thermal regulation of ocean currents in the North Atlantic. The so-called North Atlantic Oscillations (NAO) are one of the main modes of variability in the atmospheric circulation of the northern hemisphere [34]. NAO are known to significantly affect changes of climate parameters and weather conditions and, as a result, on the groundwater regime. The periodicity of NAO was not the subject of our study, but since SAT changes are well-governed by NAO [35], we can assume that there is the same cyclicity for NAO as for SAT.

These indicators are most important in winter, as they most significantly control the climate of the northern hemisphere, in particular the inter-decade variability [4]. The value of the NAO index is determined by the difference between the normalized pressure of the sea level over Gibraltar and the same indicator over southwest Iceland [34].

It is established that an abnormally dry hydrological year in the Danube basin occurs 5–6 years after an extremely low value of the NAO index (positive relationship). Thus, in December 2009–March 2010, on average, the NAO index was the lowest in almost 200 years (since 1823) at only −2.85 [4]. As we already know, the shallowest year, and not only in the basins of the Danube and Southern Bug, was 5 years later, in 2015.

It is quite logical to assume that the detected deviations from the stable rhythm are caused not by the superimposed action of several independent, more or less equivalent factors, but by the dominant action of a single factor, which is distorted by the influence of local features.

Cyclical regime-forming factors—precipitation and temperature. Thus, it turned out that the critical changes in the cyclicity of the groundwater regime mostly coincide with significant changes in temperature and precipitation in certain periods. That is, in the rhythms of regime-forming factors, in particular precipitation and temperature, after 1989, certain violations should also be sought.

The rhythms of precipitation detected by wavelet analysis (Figure 12), in contrast to the rhythms of GWT, have a slightly longer duration—8.7 years. Instead, the more usual 11-year cycles appear and disappear at the beginning (before 1980) and at the end of the period. It was noticeable that at the turn of 2002–2003, the spots of 8-year cycles were divided into smaller spots of 5–6 annual and 10–11 annual cycles.

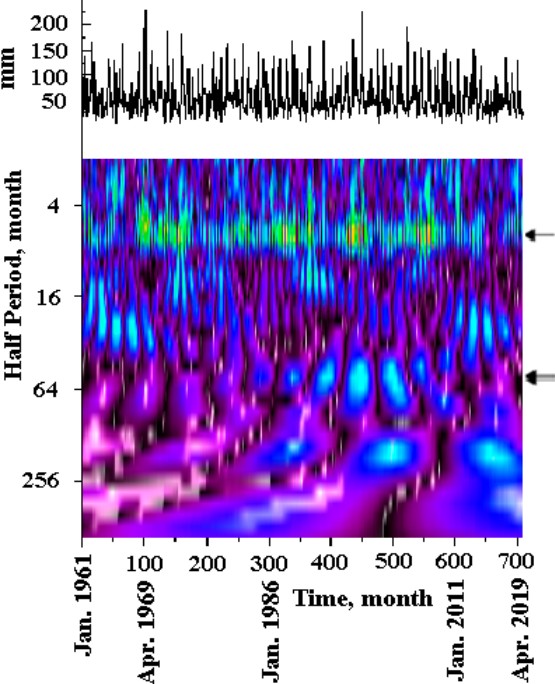

**Figure 12.** Wavelet-periodogram of long-term (1961–2019) observations of precipitation (monthly amount, mm) at the Khmilnyk weather station. Cyclicity periods: top arrow—1 year, lower arrow—8 years.

According to the results of processing the annual amounts of precipitation and total precipitation of the cold period for the near latitudes (Kamennaya Steppe, Voronezh region, Russia), a cyclic duration was revealed: 3–4, 6–8, 11, 13–16, 21–23, 26 and 34–35 years [23]. The 11-year cycles were the clearest.

It has now been proven that precipitation in the Vinnitsa region of Ukraine is affected not only by the NAO, but also by the North Sea–Caspian oscillations [36].

Another important factor traditionally associated with GWT fluctuations is the temperature of the surface air layer. Temperature fluctuations are also subject to unstable long-term cyclicity [37]. Only normal annual rhythms look stable—a strict series of narrow spots in the periodogram (Figure 13, upper arrow). According to the results of the wavelet analysis, a clear cyclicity of 8.1 + 1.1 years was also identified (Figure 13, lower arrow), which is established after 1986 and lasts until the end of the observations. For comparison, the temperature cycles previously established by the periodogram method for the Kamennaya Steppe had a duration of 3–4, 6–8, 11–12, 13–17, 19–21, 23–25 and 32–34 years [23], with the clearest cycles in 11 and 32–34 years.

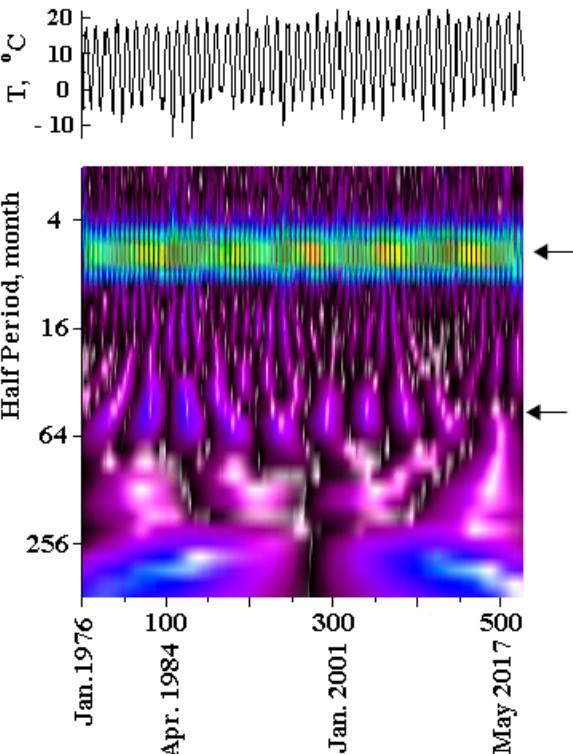

**Figure 13.** Wavelet-periodogram of long-term (1976–2019) observations of temperature cyclicity (monthly amount) at Khmilnyk city. Cyclicity periods: top arrow—1 year, lower arrow—8 years.

To determine the order of occurrence of the 8-year cycle, a longer series of temperature observations at the Khmilnyk meteorological station was analyzed. On the periodogram, which includes a number of observations of temperature at the Khmilnyk meteorological station (MS) since 1881, the 8-year cycle is only traced since 1969 (as well as on the MS "Vinnytsia"). The year 1969 marked the beginning of the phase of lowering the air temperature during the summer–autumn period [37]. In 1998, it was replaced with a warming phase that is still going on. In general, the long-term dynamics of the average monthly SAT (according to the Khmilnyk climate site) agrees well with the changes of the NAO index.

The wind speed fluctuates out of phase with the seasonal course of air temperature and has minimal values in summer. Since 1996, the wind speed has been characterized by a cyclic variability of 7.4 years. The maximum values of wind speed were observed during

1988–1993 and corresponded to the minimum monthly values of water vapor deficit. The 8-year water vapor deficit cycle has also been traced since 1988.

The most sustained cycle compatible with temperature cyclicity was the 8-year cyclicity for a number of calculated daily values of infiltration supply of groundwater in the range of 2.5–3.8 m (Figure 14). At the same time, the cyclicity of 5–6 years, which has been manifested since about 1997, is also quite pronounced. It is also noticeable that at the end of the term, after 2010, the amplitude of fluctuations in the monthly values of infiltration increases significantly.

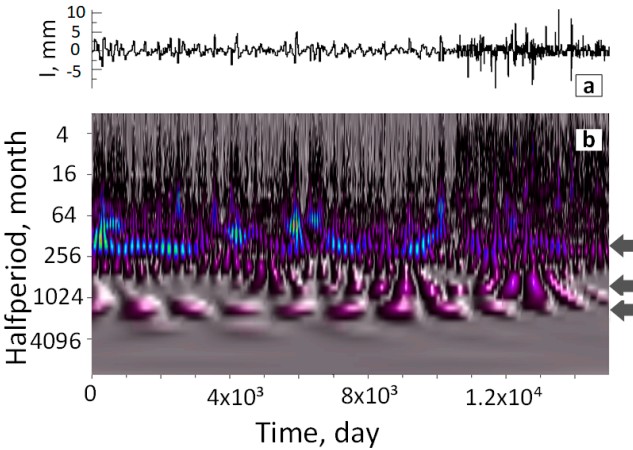

**Figure 14.** Wavelet-periodogram (**b**) of long-term (1980–2017) observations of infiltration cyclicity (day amount, mm) for GWT = 2.5–3.8 m (**a**). Cyclicity periods: top arrow—1 year, middle arrow—5–6 years, lower arrow—8 years.

Due to the study of cyclicity, it was possible to identify the subordination of turning extremes in the groundwater regime to key meteorological events [31], which are accurately interpreted as manifestations of global warming. Thus, the appearance of clear 7–8-year cycles in the oscillations of the GWT at depths of 1.5–2.0 m in 1975 is consistent with anomalous changes in temperature in this area (western Ukraine): 1974 was marked by an absolute long-term maximum air temperature in March, and in January 1975 recorded one of the largest positive deviations of the average maximum temperature (5.2 °C) [22]. Since the changes in temperature were primary (with almost constant precipitation), they have triggered changes in the value and seasonal distribution of the groundwater infiltration feed, and hence in the regime of GWT and groundwater flow to rivers. Consequently, the phase of the heightened infiltration recharge for the groundwater with GWT 1.5–2.0 m (well #5-3) began 14–15 years earlier than for depths > 3.5 m; however, a lowering of levels in this area is also traced earlier—from 1990. This was preceded by three years of maximum groundwater discharge to the river (132–160 m$^3$ year$^{-1}$ on running meter of a length) (see Figure 9), and therefore there was a preliminary depletion of the groundwater reserves in this area of catchment.

For the groundwater lying at greater depths (3.5–3.8 m), the clearest 7–8-year cycles were established later, starting from 1989, which coincides with a marked increase in winter temperatures and the first transition of average temperatures in February to positive values. The latest event marks the beginning of a period of high GWT in this area (see Figure 5). That is, in 1989, there was a certain limit value of temperature (or a set of meteorological indicators), which caused very significant changes in the regime of groundwater. Other limit dates are 2004 for groundwater close to the surface (1.0–2.0 m) and river runoff, as well as 2011–2013 for groundwater at depths of 2.7–3.8 m. With the beginning of the low-water (for surface runoff) period of 2008–2009, which is still continuing, 8-year cycles disappear, with 5–6-year ones reappearing instead. In the flow mode of the Southern Bug River, which responds more quickly to changes in precipitation, 11-year cycles can be traced back to

about 2004, and 5–6-year cycles from 2008. At the same time, there is a 5–6-year cycle for shallow (1.0–2.0 m) groundwater.

If we add the results of multiple correlation analysis, which reveals the dominant dependence of groundwater flow (at average GWT = 3.65 m) on air temperature and to a lesser extent on precipitation and surface runoff regime of the river [21] (Figure 15), it becomes obvious that the formation of 7–8-year cyclicity in the groundwater regime over the past 45 years may be fully associated with the rising temperature and its cycles. The rating of the influence of temperature on groundwater flow is the highest among other factors (weighting factor 6.06), which corresponds to the closest connection of their cyclicities, correspondingly 8.1 and 7.5–7.8 years, and since 1997 to their full synchronization. It should be noted that temperature is the main factor in the distribution of precipitation and its infiltration. As the GWT response to precipitation occurs with a significant delay (from 12 to 90 days), this explains the low statistical relationship between groundwater flow (as well as GWT) and precipitation. The priority of temperature over precipitation is explained by the greater endurance of temperatures during the period, which covers both precipitation and the reaction of GWT and groundwater flow.

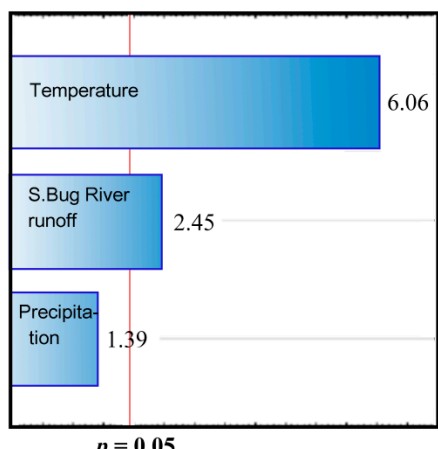

**Figure 15.** Diagram of values of standardized weights, sorted by absolute value for variables of a regression, which reveals the dependence of groundwater flow on temperature, average monthly river discharge and monthly precipitation for the period 1980–2016. Red vertical line—limit of significant coefficients (factors).

It is possible to draw the conclusion that the influence of temperature in the Forest–Steppe zone extends to depths of 3.5–4.0 m, while the critical GWT for this zone in the 1970s was 1.5–2.5 m.

There are many different periodicities in the subsoil water table fluctuations. Cyclicity shows a certain correlation with global (for example, the activity of the Sun, NAO) and local factors. The absence of obvious direct connections between phenomena (processes) does not allow us to explain the physical mechanism. In our work, we found a new type of cyclicity (8-year duration) using a more sensitive wavelet transformation method. Our attempt to find a connection with NAO or global warming is just an attempt to show that GWT is part of the global ocean–atmosphere circulation.

The close correlation between temperature variability and GWT may indicate both the effect of temperature on GWT and the existence of a common cause for their synchronous changes.

However, the dynamics of GWT and surface moisture affect the state of the surface layer of the atmosphere, and this must be taken into account when modeling the climate [38–40]. We must pay attention to the revealed phenomenon—the formation of ordered (8-yearly) fluctuations of air temperature (1969) and only later, of the same rhythms in the dynamics of the level of subsoil water (1.0–2.0 m) (1975), which, in turn, are synchronized with fluctuations in subsoil water discharge and deeper groundwater levels (1989).

Such a clear 7–8-year cycle has not yet been identified for other catchments in Ukraine. Cycles of GWT of 4, 5–6 and 12–13 years have generally been shown.

Knowledge of cyclicity in the groundwater regime, established by wavelet analysis, enables an increase in the accuracy of forecasting using observational data for time intervals with the same type of cyclicity.

## 5. Conclusions

For the first time, a clear 7–8-year cyclicity was revealed, which is characteristic both for the indicators of the groundwater regime to a depth of about 10.0 m and for the river runoff in the basin of the Southern Bug River. The 8-year cycle was confirmed in other river basins of Ukraine. In particular, in the Desna River basin, the 8-year cycle was found for GWT deeper than 4.0 m for the same period (1989–2011). This excludes its connection with factors other than global climate change. It coincides well with the cyclical changes in air temperature, but not with the cyclical nature of solar activity, which is characterized by 5–6- and 10–11-year rhythms. Wavelet analysis of the existing series of temperature data from 1881–2020 showed that 8-year cycles appeared for the first time in 140 years in 1969, which coincided with the beginning of one of the phases of temperature change in the region. After that, approximately since 1975, there is the same cyclicity in the regime of groundwater levels, successively from higher to lower levels. Moreover, 7–8-year rhythms in the fluctuation of levels and volumes of groundwater flow primarily accompany high-water periods in the groundwater regime, and are significantly weaker in low-water cycles (2008–2019). With the transition from a low-water cycle to a high-water cycle, the dominant cyclicity in the regime of groundwater and surface water changes from 5–6 years to 7–8 years. In the regime of river runoff during the low-water cycle, an 11-year cycle is added to the 5–6-year cycle.

Wavelet analysis has established that rhythms (cycles) of a certain periodicity are not constant (lasting 22 or 33 years), and the beginning of a new phase of cyclicity is associated with extreme weather events, such as the transition of the average February temperature to zero. Thus, cyclicity is a fair indicator (marker) that reveals the impact of global warming on the underground hydrosphere.

Due to the impact of global warming on the process of the terrestrial water cycle, which has been manifesting in Ukraine since 1989, the subordination of the beginning, end and duration of high-water and low-water periods in the groundwater regime do not correspond to solar activity cycles. Thus, during the 22nd solar cycle (1986–1996), which was supposed to be low-water in the study area, there was an increase in GWT and groundwater runoff. The beginning of these phenomena coincides with the consolidation of 7–8-year cyclicity in the groundwater regime at depths of 3.5–4.0 m. This is the reaction of groundwaters from outside factors, and correspondingly, the time of the beginning and end of the 7–8-year cycle depends on the GWT depth.

In the last decades, differences in solar activity have had very little effect on fluctuations in groundwater levels and consumption, in contrast to previous years, when during periods of low solar activity, the levels increased and vice versa. All of this may indicate a "restructuring" of normal cycles due to radical changes in the nature of feed (recovery of groundwater) and the shift of climatic zones to the north.

According to multiple correlation analysis, in recent decades, the temperature has become dominant in terms of the impact on the groundwater regime (with GWT from 1.5 to 4.0 m).

All of this proves that the manifestations of disturbances or changes in cyclicity in the groundwater regime may occur under the influence of global warming.

**Author Contributions:** Conceptualization, O.S. and A.S.; methodology, A.S. and O.S.; validation, N.O. and V.O. (Valeriy Osypov); formal analysis, O.S., A.S., V.G. and N.O.; investigation, O.S. and A.S.; resources, V.O. (Volodymyr Osadchyi); writing—original draft preparation, O.S.; visualization, O.S., A.S. and N.O.; funding acquisition, V.O. (Volodymyr Osadchyi). All authors have read and agreed to the published version of the manuscript.

**Funding:** This research received no external funding.

**Institutional Review Board Statement:** Not applicable.

**Informed Consent Statement:** Not applicable.

**Data Availability Statement:** Open data was not used.

**Conflicts of Interest:** The authors declare no conflict of interest.

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
