# Peer review of "Cyclicities in the Regime of Groundwater and of Meteorological Factors in the Basin of the Southern Bug River"

_water, doi:10.3390/w14142228_

Round 1

Reviewer 1 Report

Introduction

The authors apply wavelet analysis techniques to identify cyclicity in long records of shallow groundwater levels and other hydrologic variables in the southern Bug River basin, Ukraine. Their results suggest that cycles are present throughout the record and that an apparent change in cyclicity coincides with warming climates of the past two or three decades.  The manuscript is long and the writing in this revision is of variable quality; the translation is good in most areas.  The authors have provided readers with more background about the hydrology/water balance of the study area than in the original manuscript.  They also discuss, briefly, the connection between global effects such as the NAO  that could control cyclic processes in the shallow subsurface, but do not reach any firm conclusions.

The emphasis in places on possible influence of solar cycles before 1989 remains out of place and inappropriate for modern literature; the authors are reaching back for ideas from the past.  I think most modern climate and weather analyses and their own work emphasizes relations with the ocean-atmosphere system (NAO, for instance) and does not try to connect cycles to systems beyond the earth. The authors have deemphasized this approach slightly in this revised manuscript, but using text space to discuss solar influence surely takes away from the main emphasis of the paper.

General remarks

  1. The manuscript now includes a much more detailed description of the hydrologic setting, which is helpful. As I note in my marginal comments, I find it difficult to understand why groundwater levels do not closely follow the stage of the adjacent river.
  2. The manuscript focuses on wavelet analysis and now explains the technique somewhat with a figure, though the figure and explanation need to be shortened and more carefully integrated with the text.
  3. The authors still need to explain what cyclicity (if any) should be expected in GW levels over time and how they might be driven by plausible meteorologic and hydrologic processes. In most modern literature earth-system processes, particularly those involving the ocean and the tropical atmosphere, are usually identified as driving episodic/possibly cyclical change…..once direct human influences have been accounted for. Since I am not familiar with this area, I can only guess at the possible influences on river stage, and thus the shallow groundwater levels recorded by some of the wells.
  4. I am not familiar with wavelet analysis and how prone it is to show relationships that are not statistically significant or even how you decide what is significant and what is not. It would help if the authors presented a model for why there should be any periodicity (beyond annual) in GWT—wavelet analysis could then confirm the model. In North America two plausible variables—precipitation and temperature-- are linked to ocean circulation and to the tropics, but not to solar variability except on a multicentury scale.

Remarks, questions and edits are linked to the attached .pdf.

Author Response

All answers in the attached file,

With respect,
O. Shevchenko

Reviewer 2 Report

The manuscript needs improvement in terms of layout and homogeneity.

1. What will happen if we change the mother wavelet in the analysis? Does it affect the results?

2. Please concise the results. There are too much analysis without physical explnanation.

Author Response

Answers in the attached file

Sincerely,
O. Shevchenko

Reviewer 3 Report

The manuscript needs both English and formatting revisions. The locations where edits are needed are shown below:

§  Line 31-39: The “Highlights” paragraph repeating same thing from “Abstract”. Merge this paragraph with Abstract. A separate paragraph under different heading is not necessary. Please follow the article structure: “Instructions for Authors > Research Manuscript Sections”.

§  Reference style: This manuscript did not follow the journal specific style for reference. In the text, reference numbers should be placed in square brackets [ ], and placed before the punctuation; for example [1], [1–3] or [1,3]. References must be numbered in order of appearance in the text, not in alphabetic order.

§  Font type/size is not consistent throughout the manuscript. Use journal specific style.

§  Line 93: “Research methods” should be number- 2

§  Line 102: Meaning is not clear, rewrite this sentence.

§  There are lots of typing mistakes, i.e., line- 141, 143, 145, 148 and so on.

§  Figure 1 (a, b, c): The fonts (i.e., signal, make?) are not clear. Make the figures large enough so that the figure is easily readable.

§  Line 206 – 207: Add a map/figure to show the relative locations of the wells. It is difficult for readers to visualize the well locations from text. Although figure 2 talks about wells but it is not clear.

§  Figure 2: Fonts are too small in the figure (i.e., latitude/longitude, legend, scale, basin names).

§  Figure 3: In the x-axis, replace the numbers with month name (i.e., Jan, Feb).

§  In all the numbers, replace comma (,) with decimal (.) symbol in text as well as figures.

§  Line 255, 281, 300, 425, 480, 513: Make all the sub-headings bold.

§  Line 316: You have used different name to represent same well (i.e., well 1 & well #5-3 for same well) which is confusing. Be consistent everywhere.

§  Figure 4: In the legend instead of only numbers 1, 2, mention what it is (i.e., well 1). 

§  Figure 5, 6, 7, 10, 11, 12, 13, 14: Make the figures large and increase resolution.

§  Line 724: This reference was not used anywhere in the text. Reference needs to be used in text as well as in the reference list.

Author Response

Answers in the attached file

Sincerely,
O. Shevchenko et al.

Round 2

Reviewer 2 Report

It can be accepted now.